# Semi-Supervised Learning Based Cascaded Pocket U-Net for Organ and Pan-cancer Segmentation in Abdomen CT

Tao Wang[1(✉)], Xiaoling Zhang[1], Wei Xiong[1], Shuoling Zhou[1], and Xinyue Zhang[1]

School of Biomedical Engineering, Southern Medical University, Guangzhou 510515, China
wangtao_9802@sina.cn

**Abstract.** In clinical practice, CT scans are frequently employed as the primary imaging modality for detecting prevalent tumors arising from the abdominal organs. Hence, the accomplishment of simultaneous organ segmentation and pan-cancer segmentation in abdominal CT scans holds significant importance in decreasing the workload of clinical practitioners. To maximize the utilization of partially labeled and unlabeled data, a iterative training strategy through a semi-supervised approach based on pseudo labels is employed in this work. Furthermore, to reduce parameter size of model and increase efficiency of GPU utilization, the proposed method is built upon the pocket U-Net architecture. The methodology involves a cascaded network consisting of two parts: initially, a segmentation network trained on labeled data refines the low-resolution pocket U-Net to reduce image dimensions; subsequently, the high-resolution pocket U-Net conducts intricate segmentation to precisely delineate organ and tumor regions. As demonstrated by the evaluation outcomes on the FLARE 2023 validation dataset, the proposed method achieves an average dice similarity coefficient (DSC) of 88.94% for organs and 15.92% for tumors, along with normalized surface dice (NSD) values of 93.31% for organs and 0.0816% for tumors, with minimal parameter size. Furthermore, the average inference time is 82.61 seconds, with an average maximum GPU memory usage of 3560M. Codes are available at https://github.com/wt812549723/FLARE2023_solution.

**Keywords:** Organ and pan-cancer segmentation · Semi-supervised learning · Minimal parameter size

## 1 Introduction

Accurate and fast segmentation of abdominal organs and pan-cancer in abdominal CT scans is crucial to reduce the clinician's workload and improve the efficiency of diagnosis and treatment. However, abdominal organ and pancancer segmentation faces several challenges: (1) Obtaining labels is both time-consuming and labor-intensive. (2) A significant amount of unlabeled and par-

tially labeled data is available to improve segmentation performance. (3) Balancing segmentation performance, rapid inference speed, and efficient GPU utilization. (4) The segmentation performance of certain organs and pan-cancers is limited by the variations in size and morphology among different organs and the morphological differences and heterogeneity of tumors within various organs.

As a result of the unlabeled and partially labeled data available, the proposed method adopts a semi-supervised learning framework. Semi-supervised methods can be categorized into three groups: pseudo-label-based [5,10], consistency-based [3,11], and hybrid methods [19]. Among these methods, pseudo-label based methods are often devoid of introducing supplementary parameters and burdens to the model. In consideration of the model size and inference speed, the proposed method incorporates a pseudo-label based semi-supervised learning framework. Moreover, many existing medical segmentation models have been extended upon the foundation of nnU-Net, which has proven to be an excellent solution capable of addressing a variety of medical segmentation tasks [9]. However, the default configuration of nnU-Net employs the traditional U-Net architecture, often leading to concerns about large model parameter sizes and slow inference speed. Furthermore, nnU-Net was originally designed for fully supervised segmentation tasks, necessitating extensions to incorporate aspects of semi-supervised learning.

Therefore, a semi-supervised learning based cascaded pocket U-Net is proposed to achieve abdominal multi-organ and pan-cancer segmentation. First, the proposed method builds on the nnU-Net framework and extends it to introduce a pseudo-label-based semi-supervised learning strategy through iterative training. This strategy effectively utilizes a substantial amount of unlabeled and partially labeled data. In addition, the use of the pocket U-Net reduces the network parameters, thereby decreasing the GPU utilization and increasing the compatibility with a wide range of devices. Subsequently, a cascaded network is employed to accelerate segmentation: the first-tier network performs a region-of-interest (ROI) segmentation to reduce image dimensions, followed by the second-tier network to perform a refined segmentation. Finally, the proposed method achieves efficient segmentation of abdominal organs and pan-cancer.

The main contributions of the proposed method are as follows:

– We integrated a semi-supervised training strategy based on pseudo labels into the nnU-Netv2 framework.

– We implemented a two-stage cascaded architecture to enhance the inference speed.

– We employed the Pocket U-Net architecture as the backbone network, resulting in a significant reduction in the model's parameter size. This optimization ensures efficient GPU utilization.

## 2 Method

### 2.1 Preprocessing

The preprocessing steps in our proposed method align with the approach for handling CT data as defined by nnU-Net [9]. These steps encompass the following procedures: (a) the exclusion of irrelevant background regions through cropping; (b) the application of CT value truncation to eliminate superfluous information; (c) the utilization of mean and standard deviation computed from all training samples for normalization; (d) the resampling of all images to ensure a consistent target (targets are set to 4.0mm×1.2mm×1.2mm and 2.5mm×0.8mm×0.8mm for ROI segmentation network and detail segmentation network in the proposed method, respectively).

### 2.2 Proposed Method

**Pocket U-Net** As shown in Fig. 1, traditionally, in the U-Net architecture, it is customary to double the number of feature channels following each downsampling operation. However, this practice significantly contributes to the increase in the parameter size of model. Moreover, previous research in medical image segmentation has demonstrated that controlling the expansion of feature channels can maintain satisfactory performance while effectively managing parameter size [2]. Inspired by this study, a specialized variant of U-Net, named as Pocket U-Net, was introduced as the backbone network. The difference between the traditional U-Net and Pocket U-Net is illustrated in Fig. 1, wherein Pocket U-Net maintains a consistent number of feature channels across all scales, thereby ensuring optimal GPU utilization.

**ROI Segmentation Network** Owing to the localization of major abdominal organs within a specific region, known as the ROI, voxels outside the ROI introduce additional complexity during both model training and inference. Thus, the first part of the cascaded network is designed with an ROI segmentation network to identify the ROI where abdominal organs are present, thereby reducing computational overhead. Specifically, all organ and tumor regions are set as foreground regions of the same labels (all labels are set to 1 in our method), and a pocket U-Net is employed to identify all foreground regions. Considering the relatively straightforward nature of this task and its low precision requirements, a light pocket U-Net was trained using only fully labeled data.

Herein, a light Pocket U-Net implies smaller target spacing (i.e., reduced image dimensions), small patch sizes, shallow depth, and multiple downsampling operations in z-axis.

**Detail Segmentation Network** Once the ROI has been delineated by the ROI segmentation network, the image is cropped based on the ROI. Subsequently, the cropped image is then input to the detail segmentation network. In contrast

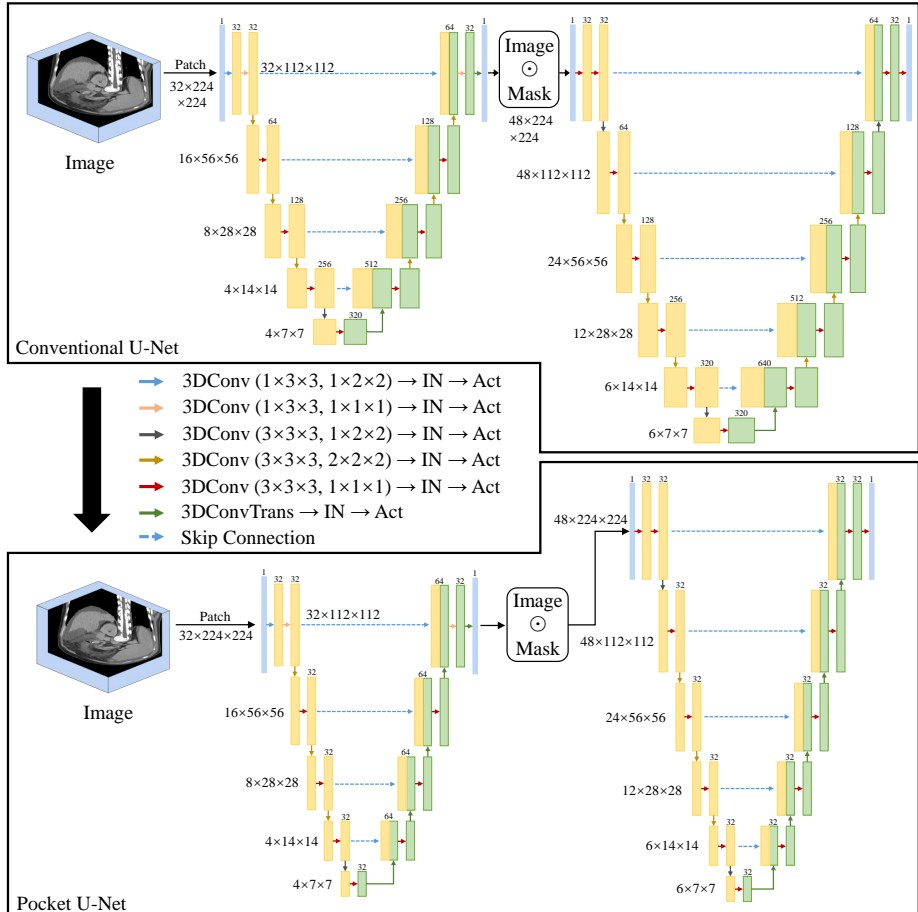

**Fig. 1.** Network Architecture. In contrast to the conventional U-Net, Pocket U-Net primarily modifies the number of channels, resulting in a substantial reduction of model parameters.

to the ROI segmentation network, the detail segmentation network utilizes a larger target spacing to retain finer details in the image. Furthermore, the detail segmentation network employs larger patch sizes and reduces the frequency of downsampling operations, facilitating the network in capturing both fine-grained details and contextual information. Another notable difference from the ROI segmentation network is that while the ROI segmentation network is trained once using fully labeled data, the detail segmentation network engages in semi-supervised learning and undergoes multiple iterations of training utilizing all available data.

**Semi-supervised learning** Semi-supervised learning plays a pivotal role in effectively harnessing partially labeled and unlabeled data [18,19]. The specific implementation of semi-supervised learning can be illustrated by the detail segmentation network, which involves a five-step process. First, the detail segmentation network undergoes initial training using fully labeled images. Following this initial training phase, the network is then deployed to generate initial pseudo labels for both partially labeled and unlabeled data. Second, in the second round of training for the detail segmentation network, in addition to the fully labeled data, partial labeling and unlabeled data are incorporated. Notably, a additional process is applied to the partially labeled data in this round. Specifically, the unlabeled portions within the partially labeled data are supplemented with the generated initial pseudo labels, whereas the labeled portions remain unaltered to ensure label accuracy. Meanwhile, the unlabeled data are entirely assigned pseudo labels. Similarly, the detail segmentation network trained in the second round is utilized to generate new pseudo labels. Third, building upon the pseudo labels generated in the second round, the procedure described in the second step is reiterated to conduct a third round of training. Furthermore, the detail segmentation network obtained from the third round of training can likewise be employed for pseudo label generation. Fourth, relying on the pseudo labels derived from the first three rounds of training, partially labeled and unlabeled data with pseudo labels for the final training are selected based on computed uncertainty scores [8]. Specifically, uncertainty scores are calculated by measuring the average overlap between the pseudo labels obtained in the first and second rounds and between the pseudo labels obtained in the second and third rounds. A higher degree of overlap corresponds to a lower uncertainty score, and the mathematical formula for calculating these scores is expressed as follows:

$$us = \frac{1}{3} \sum_{i=1}^{3} \frac{\mathbf{SUM}(\mathbf{v_i^{x,y,z}} \neq \mathbf{v_{i-1}^{x,y,z}})}{\mathbf{SUM}(\mathbf{v_i^{x,y,z}} > \mathbf{0})} \tag{1}$$

where $us$ denotes the uncertainty scores, $v_i^{x,y,z}$ denotes the value of voxels with coordinates $(x, y, z)$ in the pseudo label obtained through $i$th round of training, and $\mathbf{SUM}(\cdot)$ denotes the sum of the number of voxels that meet the condition. Based on a pre-defined uncertainty score threshold, a selection was made to include partially labeled and unlabeled data for the final training. Additionally,

the pseudo labels generated in the third round are utilized as labels for the unlabeled portions of these data, thus converting them into fully labeled data. Fifth and finally, the selected partially labeled and unlabeled data, now equipped with pseudo labels, are used in conjunction with fully labeled data for the final training of the detail segmentation network. The final trained detail segmentation network is seamlessly integrated with the ROI segmentation network to form the ultimate model.

**Loss Function** Given the iterative training strategy used in semi-supervised learning, there is no requirement to introduce an additional loss specifically for semi-supervised learning purposes. Herein, a combination of the cross-entropy loss and Dice loss was utilized. This loss functions are presently well-established and extensively applied in the field of medical image segmentation [9,11].

### 2.3 Post-processing

To prevent isolated errors in the foreground regions, a connected component analysis was employed for segmentation result of each organ. Specifically, only the largest connected component was retained to improve performance. Additionally, since the images are cropped between the ROI segmentation network and the detail segmentation network, it is necessary to reconstruct the images in post-processing based on the cropping coordinates.

## 3 Experiments

### 3.1 Dataset and evaluation measures

The FLARE 2023 challenge is an extension of the FLARE 2021-2022 [13][14], aiming to aim to promote the development of foundation models in abdominal disease analysis. The segmentation targets cover 13 organs and various abdominal lesions. The training dataset is curated from more than 30 medical centers under the license permission, including TCIA [4], LiTS [1], MSD [17], KiTS [6,7], and AbdomenCT-1K [15]. The training set includes 4000 abdomen CT scans where 2200 CT scans with partial labels and 1800 CT scans without labels. Among them, the 2200 partially labeled CT scans contain partially labeled data for all organs and tumors, i.e., fully labeled data. The validation and testing sets include 100 and 400 CT scans, respectively, which cover various abdominal cancer types, such as liver cancer, kidney cancer, pancreas cancer, colon cancer, gastric cancer, and so on. The organ annotation process used ITK-SNAP [20], nnU-Net [9], and MedSAM [12].

The evaluation metrics encompass two accuracy measures—Dice Similarity Coefficient (DSC) and Normalized Surface Dice (NSD)—alongside two efficiency measures—running time and area under the GPU memory-time curve. These metrics collectively contribute to the ranking computation. Furthermore, the running time and GPU memory consumption are considered within tolerances of 15 seconds and 4 GB, respectively.

### 3.2 Implementation details

**Environment settings** The development environments and requirements are presented in Table 1.

**Table 1.** Development environments and requirements.

| | |
|---|---|
| System | Ubuntu 20.04.6 LTS |
| CPU | Intel(R) Xeon(R) Gold 5120 CPU @ 2.20GHz |
| RAM | 16×4GB; 2.67MT/s |
| GPU (number and type) | One GeForce RTX 2080 Ti 11G |
| CUDA version | 11.7 |
| Programming language | Python 3.9 |
| Deep learning framework | torch 2.0.1 |
| Code | https://github.com/wt812549723/FLARE2023_solution |

**Training protocols** The handling of unlabeled images and partially labeled data has been comprehensively explained in Section 2.3. Additionally, the data augmentation techniques employed in our proposed method align with the default settings utilized in nnU-Net [9], encompassing rotations, scaling, Gaussian noise, Gaussian blur, adjustments in brightness and contrast, simulation of low resolution, gamma correction, and mirroring. Furthermore, both the patch sampling strategy and the criteria for optimal model selection are entirely in accordance with the guidelines established by nnU-Net [9]. To further enhance segmentation efficiency, testtime augmentation (TTA) has been disabled during inference, and the step size for sliding window prediction has been set to 1.

**Table 2.** Training protocols of ROI segmentation network.

| | |
|---|---|
| Network initialization | "He" normal initialization |
| Batch size | 2 |
| Patch size | 32×224×224 |
| Total epochs | 1000 |
| Optimizer | SGD with nesterov momentum ($\mu = 0.99$) |
| Initial learning rate (lr) | 0.01 |
| Lr decay schedule | halved by 200 epochs |
| Training time | 35.3 hours |
| Loss function | Cross-Entropy loss + Dice loss |
| Number of model parameters | 5.76M[1] |
| Number of flops | 485.17G[2] |

**Table 3.** Training protocols for detail segmentation network.

| | |
|---|---|
| Network initialization | "He" normal initialization |
| Batch size | 2 |
| Patch size | 48×224×224 |
| Total epochs | 1000 |
| Optimizer | SGD with nesterov momentum ($\mu = 0.99$) |
| Initial learning rate (lr) | 0.01 |
| Lr decay schedule | halved by 200 epochs |
| Loss function | Cross-Entropy loss + Dice loss |
| Training time | 44.0 hours |
| Number of model parameters | 7.97M[3] |
| Number of flops | 2.77T [4] |

# 4 Results and discussion

**Table 4.** Quantitative evaluation results.

| Target | Public Validation | | Online Validation | | Testing | |
|---|---|---|---|---|---|---|
| | DSC(%) | NSD(%) | DSC(%) | NSD(%) | DSC(%) | NSD (%) |
| Liver | 98.12 ± 0.83 | 98.71 ± 1.82 | 97.97 | 98.38 | 75.37 | 75.40 |
| Right Kidney | 88.94 ± 21.75 | 89.54 ± 21.63 | 90.42 | 90.78 | 73.35 | 73.59 |
| Spleen | 94.38 ± 11.86 | 94.96 ± 11.89 | 93.91 | 94.84 | 72.70 | 73.34 |
| Pancreas | 85.72 ± 5.73 | 95.84 ± 3.21 | 84.17 | 94.65 | 68.77 | 74.35 |
| Aorta | 96.69 ± 3.19 | 98.46 ± 3.87 | 96.91 | 98.67 | 76.67 | 77.95 |
| Inferior vena cava | 92.32 ± 4.67 | 93.03 ± 5.44 | 91.90 | 92.46 | 72.51 | 73.35 |
| Right adrenal gland | 88.64 ± 4.71 | 97.34 ± 2.64 | 88.22 | 97.07 | 66.57 | 73.17 |
| Left adrenal gland | 84.69 ± 9.32 | 93.82 ± 8.03 | 84.31 | 93.22 | 66.47 | 72.77 |
| Gallbladder | 81.20 ± 26.12 | 81.33 ± 27.32 | 82.53 | 82.65 | 62.37 | 63.07 |
| Esophagus | 80.21 ± 16.40 | 89.39 ± 15.51 | 81.71 | 91.19 | 68.28 | 74.51 |
| Stomach | 92.12 ± 6.40 | 94.61 ± 7.12 | 92.43 | 94.75 | 72.16 | 73.62 |
| Duodenum | 80.58 ± 8.53 | 92.68 ± 6.22 | 80.54 | 92.52 | 65.21 | 73.27 |
| Left kidney | 90.27 ± 16.67 | 91.42 ± 15.77 | 91.26 | 91.82 | 72.47 | 73.00 |
| Tumor | 16.27 ± 21.01 | 8.87 ± 17.54 | 15.92 | 8.16 | 15.67 | 0.06 |
| Average | 88.76 ± 15.77 | 93.16 ± 15.77 | 88.94 | 93.31 | 70.22 | 73.18 |

## 4.1 Quantitative results on validation set

Based on the results presented in Table 4, our method achieved notable performance in organ segmentation on the publicly available validation dataset. Specifically, it attained an average DSC of 88.76% and an average NSD of 93.16%. However, for tumor segmentation on the same dataset, our method achieved a

**Table 5.** Comparison results on the public validation set, where "w/o" denotes "without".

| Target | The Proposed Method | | Pocket U-Net (w/o unlabeled data) | |
|---|---|---|---|---|
| | DSC(%) | NSD(%) | DSC(%) | NSD(%) |
| Liver | 98.12 ± 0.83 | 98.71 ± 1.82 | 96.14 ± 4.87 | 95.78 ± 5.89 |
| Right Kidney | 88.94 ± 21.75 | 89.54 ± 21.63 | 90.24 ± 19.63 | 89.64 ± 20.27 |
| Spleen | 94.38 ± 11.86 | 94.96 ± 11.89 | 80.42 ± 24.33 | 78.94 ± 23.35 |
| Pancreas | 85.72 ± 5.73 | 95.84 ± 3.21 | 84.32 ± 8.80 | 94.91 ± 6.92 |
| Aorta | 96.69 ± 3.19 | 98.46 ± 3.87 | 96.41 ± 3.79 | 97.89 ± 4.80 |
| Inferior vena cava | 92.32 ± 4.67 | 93.03 ± 5.44 | 90.93 ± 7.29 | 91.03 ± 8.60 |
| Right adrenal gland | 88.64 ± 4.71 | 97.34 ± 2.64 | 86.44 ± 7.05 | 95.79 ± 5.32 |
| Left adrenal gland | 84.69 ± 9.32 | 93.82 ± 8.03 | 85.10 ± 8.30 | 94.03 ± 6.73 |
| Gallbladder | 81.20 ± 26.12 | 81.33 ± 27.32 | 73.26 ± 27.15 | 72.43 ± 27.92 |
| Esophagus | 80.21 ± 16.40 | 89.39 ± 15.51 | 80.56 ± 17.45 | 90.11 ± 16.71 |
| Stomach | 92.12 ± 6.40 | 94.61 ± 7.12 | 87.13 ± 13.60 | 89.64 ± 14.32 |
| Duodenum | 80.58 ± 8.53 | 92.68 ± 6.22 | 79.76 ± 10.87 | 92.90 ± 7.28 |
| Left kidney | 90.27 ± 16.67 | 91.42 ± 15.77 | 88.67 ± 19.06 | 87.46 ± 20.53 |
| Tumor | 16.27 ± 21.01 | 8.87 ± 17.54 | 39.26 ± 29.31 | 26.77 ± 22.11 |
| Average | 88.76 ± 15.77 | 93.16 ± 15.77 | 86.11 ± 6.37 | 90.04 ± 6.89 |

comparatively lower average DSC of 16.27% and a moderate average NSD of 8.87%. Furthermore, on the online validation dataset, our method consistently demonstrated strong performance in organ segmentation, with an average DSC of 88.76% and an average NSD of 93.16%. However, in the challenging task of tumor segmentation on this dataset, our method achieved an average DSC of 15.92 and an average NSD of 8.16%. In addition, the post-processing method based on the largest connected analysis method brought an improvement of 1.21% to the model.

Analyzing these results reveals valuable insights. Our proposed method excels in segmenting larger organs, such as the liver, spleen, and stomach, as well as in delineating major blood vessels like the aorta and inferior vena cava. In contrast, its performance appears less robust when applied to smaller organs such as the gallbladder, esophagus, and duodenum. Notably, our method faces challenges in tumor segmentation, likely attributed to the diverse nature of tumors, their widespread distribution, and the absence of distinct concentration zones.

The comparison results indicate that the inclusion of unlabeled data has yielded a favorable impact on organ segmentation. Surprisingly, however, unlabeled data has adversely affected tumor segmentation, resulting in a significant decline in tumor segmentation metrics. This phenomenon may be attributed to the relatively low accuracy of tumor pseudo-labels, which introduced additional noise into the model.

**Table 6.** Quantitative evaluation of segmentation efficiency in terms of the running them and GPU memory consumption. Total GPU denotes the area under GPU Memory-Time curve. Evaluation GPU platform: NVIDIA QUADRO RTX5000 (16G).

| Case ID | Image Size | Running Time (s) | Max GPU (MB) | Total GPU (MB) |
|---|---|---|---|---|
| 0001 | (512, 512, 55) | 54.24 | 3410 | 72616 |
| 0051 | (512, 512, 100) | 66.54 | 3710 | 94022 |
| 0017 | (512, 512, 150) | 92.46 | 3776 | 141352 |
| 0019 | (512, 512, 215) | 100.37 | 3546 | 148469 |
| 0099 | (512, 512, 334) | 117.00 | 3706 | 169899 |
| 0063 | (512, 512, 448) | 150.09 | 3770 | 224764 |
| 0048 | (512, 512, 499) | 169.41 | 3746 | 263733 |
| 0029 | (512, 512, 554) | 221.69 | 3960 | 392044 |

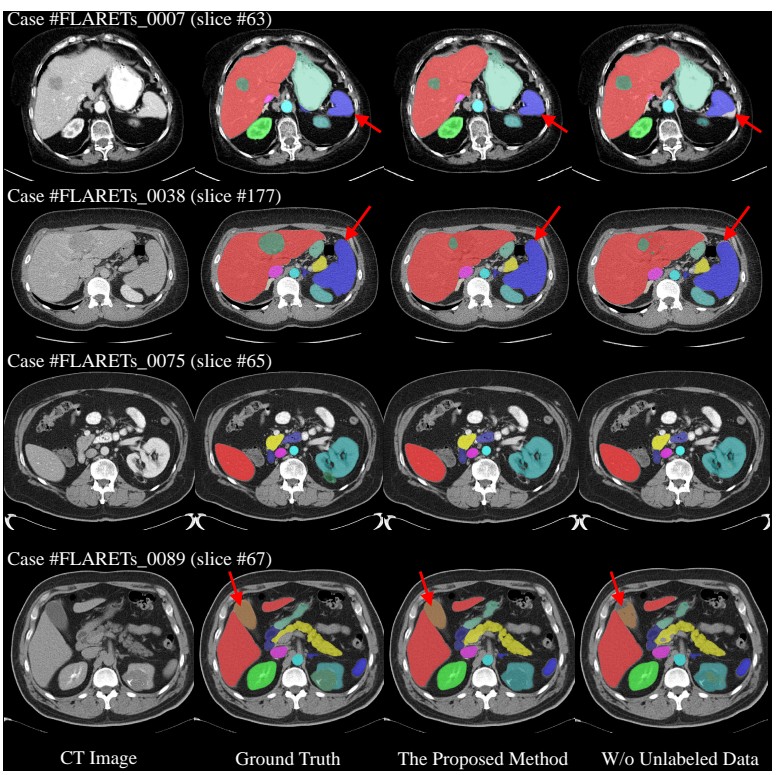

**Fig. 2.** Two examples with good segmentation results (the above two rows) and two examples with bad segmentation results (the following two rows) in the validation set. Among these, the first column represents the original images, the second column shows the gold standard, the third column displays the results of our proposed method, and the fourth column demonstrates the outcomes of the method that does not utilize unlabeled data. The red arrow indicates the improvement of semi-supervised training.

### 4.2 Qualitative results on validation set

Based on the results shown in Figure 2, we have observed that our approach is not particularly sensitive to low-contrast tumor segmentation. The proposed method tends to classify low-contrast tumors as normal regions or background. Furthermore, while models that do not utilize unlabeled data perform significantly better than the proposed method in terms of tumor segmentation metrics, they also exhibit inaccuracies in tumor segmentation. In organ segmentation, the introduction of unlabeled data has led to performance improvements. It can be observed that our method performs better in spleen segmentation for Case #FLARETs_0007 and Case #FLARETs_0038, as well as gallbladder segmentation for Case #FLARETs_0089. Hence, the primary limitation of our approach lies in tumor segmentation.

### 4.3 Segmentation efficiency results on validation set

The segmentation efficiency results for the validation dataset are presented in Table 5. These primarily include running time, GPU memory consumption, and the area under the GPU Memory-Time curve. The shortest running time was 43.53 seconds, the longest was 221.69 seconds, with an average of 82.61 seconds. GPU memory consumption ranged from a minimum of 3200MB to a maximum of 4388MB, averaging at 3560MB. The area under the GPU Memory-Time curve varied from a minimum of 53508MB to a maximum of 392044MB, with an average of 120090MB. Compared to traditional U-Net, our approach offers significant advantages in terms of parameter size. In the ROI segmentation network, the parameter size for the traditional U-Net constructed by nnU-Net is 123.61M, whereas the parameter size of our method is only 5.76M. As for detail segmentation network, the parameter sizes of traditional U-Net and our method are 235.60M and 7.97M, respectively.

### 4.4 Results on final testing set

As shown in Table 4, our method consistently demonstrated perform an average DSC of 88.76% and an average NSD of 93.16% in organ segmentation. In the challenging task of tumor segmentation on this dataset, our method achieved an average DSC of 15.92% and an average NSD of 8.16%.

### 4.5 Limitation and future work

The current method exhibits notable limitations in tumor segmentation, particularly in the identification of widely distributed and variably-sized abdominal tumors. In essence, the challenge of pan-cancer segmentation persists. Moreover, there is ample room for enhancing the model's efficiency. Despite the significant advantage in terms of model parameters, several areas can still be fine-tuned to further reduce inference time. Furthermore, there is a pressing need to explore novel semi-supervised learning approaches to fully exploit the potential of unlabeled data in tumor segmentation.

# 5 Conclusion

While our proposed method has demonstrated promising results in organ segmentation, it encounters substantial challenges in the realm of tumor segmentation. Surprisingly, the integration of unlabeled data had a detrimental impact on tumor segmentation. Furthermore, despite the minimal parameter count of our method, there is potential for further enhancement in segmentation efficiency across various aspects.

**Acknowledgements** The authors of this paper declare that the segmentation method they implemented for participation in the FLARE 2023 challenge has not used any pre-trained models nor additional datasets other than those provided by the organizers. The proposed solution is fully automatic without any manual intervention. We thank all the data owners for making the CT scans publicly available and CodaLab [16] for hosting the challenge platform.

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

**Table 7.** Checklist Table. Please fill out this checklist table in the answer column.

| Requirements | Answer |
|---|---|
| A meaningful title | Yes |
| The number of authors ($\leq 6$) | Number 5 |
| Author affiliations, Email, and ORCID | Yes |
| Corresponding author is marked | Yes |
| Validation scores are presented in the abstract | Yes |
| Introduction includes at least three parts: background, related work, and motivation | Yes |
| A pipeline/network figure is provided | Figure number 1 |
| Pre-processing | Page number 3 |
| Strategies to use the partial label | Page number 5 |
| Strategies to use the unlabeled images. | Page number 5 |
| Strategies to improve model inference | Page number 4 |
| Post-processing | Page number 5 |
| Dataset and evaluation metric section is presented | Page number 8 |
| Environment setting table is provided | Table number 1 |
| Training protocol table is provided | Table number 2/3 |
| Ablation study | Page number 5 |
| Efficiency evaluation results are provided | Table number 6 |
| Visualized segmentation example is provided | Figure number 2 |
| Limitation and future work are presented | Yes |
| Reference format is consistent. | Yes |