# OpenReview forum: "Semi-Supervised Learning Based Cascaded Pocket U-Net for Organ and Pan-cancer Segmentation in Abdomen CT"
_MICCAI.org/2023/FLARE — Submitted to FLARE 2023_

### Official Review · Reviewer_r5Hb · 2023-09-27
**The contents are complete, but some details need to be added.**

**Rating:** 7
**Confidence:** 5

**Review:**

Summary:

This paper introduces a strategy for pseudo-label-based semi-supervised iterative training and optimization of the U-Net network, aiming to enhance the efficiency of model inference. Furthermore, a cascade network is constructed based on the Pocket U-Net architecture, enabling a two-stage method for segmentation of organs and tumor regions.

Comments:

1.The experiment would have been more comprehensive if it included additional details about both the Conventional U-Net and Pocket U-Net models, including a thorough comparison of their efficiency.
2.This article contains some spelling errors, such as the incorrect labeling of "Figure X" in section 2.2.
3.The paper does not explicitly demonstrate the specific improvement effect of utilizing the largest connected component.

---

> ### Author Response · Authors · 2023-11-10
> **Response**
>
> Thank you for your comments.
> In Section 4.3, we have introduced detailed explanations highlighting the efficiency advantages of our method, particularly concerning parameter size of model. Concurrently, we have rectified the previously mentioned typographical errors. Finnaly, we expounded on the enhancements facilitated by the largest connected component method in Section 4.1.

---

### Official Review · Reviewer_5TLd · 2023-10-04
**Good, but needs more detail**

**Rating:** 6
**Confidence:** 4

**Review:**

- Pros
    1. Introduced a two-stage cascaded network called Pocket U-net and an iterative training strategy.
- Cons
    1. Abstract: The DSC and NSD performance metrics lack a "%" symbol, and there's no mention of efficiency performance in terms of inference time and GPU usage.
    2. Fig 1: The font size is small, lacks a description, and contains placeholder/template content.
    3. Methods: There's no detailed information about the dataset used to train the two networks, especially it's unclear what the term "fully labeled data" refers to.
    4. There are typographical errors, such as Pocked U-Net(Table 5), smallr and Fig. X.
    5. The performance on tumors is somewhat low.

---

> ### Author Response · Authors · 2023-11-10
> **Response**
>
> Thank you for your comments.
> The content of Figure 1 has been revised for increased clarity. Explanations have been added to provide better context and understanding. Meanwhile, percentage symbols (%) have been added after all DSC and NSD indicators, contributing to a clearer representation of these metrics. Moreover, in Section 3.1, detailed explanations have been incorporated to clarify the concept of fully labeled data, contributing to a more comprehensive understanding. The previously mentioned typographical errors have been identified and corrected for accuracy. Finally, Section 4.1 includes an analysis that explores potential reasons for poor tumor response.

---

### Official Review · Reviewer_YUsJ · 2023-10-04
**Good approach, but needs more explanation and analysis.**

**Rating:** 6
**Confidence:** 4

**Review:**

- Summary
    - Proposed the ROI segmentation network, detail segmentation network and a semi-supervised learning approach.
- Comments
    - The description of the training strategy using partially labeled and unlabeled data is not adequately detailed. Highlighted indications in Fig. 2 would be beneficial. Lack of analysis for cases where segmentation was difficult.

---

> ### Author Response · Authors · 2023-11-10
> **Response**
>
> Thank you for your comments.
> We have detailed the strategies for utilizing partially labeled data (including fully labeled data) and unlabeled data in the 'Semi-supervised learning' subsection in Section 2.2. To better visualize the positive impact of unlabeled data, we have revised Fig. 2. The primary challenge in our method lies in the tumor segmentation task, as evident in Fig. 2. Section 4.1 provides a detailed analysis of this difficulty.

---

### Decision · Program_Chairs · 2023-10-24

Accept